# FXR Agonism with Bile Acid Mimetic Reduces Pre-Clinical Triple-Negative Breast Cancer Burden

**DOI:** 10.3390/cancers16071368

**Published:** 2024-03-30

**Authors:** Sydney C. Joseph, Samson Eugin Simon, Margaret S. Bohm, Minjeong Kim, Madeline E. Pye, Boston W. Simmons, Dillon G. Graves, Stacey M. Thomas-Gooch, Ubaid A. Tanveer, Jeremiah R. Holt, Suriyan Ponnusamy, Laura M. Sipe, D. Neil Hayes, Katherine L. Cook, Ramesh Narayanan, Joseph F. Pierre, Liza Makowski

**Affiliations:** 1Department of Medicine, Division of Hematology and Oncology, College of Medicine, The University of Tennessee Health Science Center, Memphis, TN 38163, USA; 2Department of Microbiology, Immunology and Biochemistry, College of Medicine, The University of Tennessee Health Science Center, Memphis, TN 38163, USA; 3Department of Biological Sciences, University of Mary Washinton, Fredericksburg, VI 22401, USA; 4UTHSC Center for Cancer Research, College of Medicine, The University of Tennessee Health Science Center, Memphis, TN 38163, USA; 5Department of Cancer Biology, Wake Forest University School of Medicine, Winston Salem, NC 27157, USA; klcook@wakehealth.edu; 6Department of Nutritional Sciences, College of Agricultural and Life Sciences, University of Wisconsin-Madison, Madison, WI 53706, USA

**Keywords:** nuclear hormone receptor, agonist, bile acid receptor, FXR, TGR5, triple-negative breast cancer, TNBC

## Abstract

**Simple Summary:**

Breast cancer (BC) is the most common malignancy and the leading cause of cancer mortality in women. It is possible that signals arising from the gut microbiome are protective in BC. We used common drugs designed to turn on pathways to mimic one major signal coming from gut microbes, namely bile acids. Taking advantage of a drug called Ocaliva, which is commercially available for another disease, we repurposed this drug to test its anti-cancer potential. Ocaliva worked on a specific protein target called FXR to reduce tumor growth and killed cancer cells. In contrast, a similar drug that targets a different protein called TGR5 was not effective in killing cancer cells. Taken together, findings suggest that using Ocaliva or other bile acid signals derived from the gut microbiome to target the protein FXR could be an important new therapeutic strategy for individuals with aggressive BC.

**Abstract:**

Bariatric surgery is associated with improved outcomes for several cancers, including breast cancer (BC), although the mechanisms mediating this protection are unknown. We hypothesized that elevated bile acid pools detected after bariatric surgery may be factors that contribute to improved BC outcomes. Patients with greater expression of the bile acid receptor FXR displayed improved survival in specific aggressive BC subtypes. FXR is a nuclear hormone receptor activated by primary bile acids. Therefore, we posited that activating FXR using an established FDA-approved agonist would induce anticancer effects. Using in vivo and in vitro approaches, we determined the anti-tumor potential of bile acid receptor agonism. Indeed, FXR agonism by the bile acid mimetic known commercially as Ocaliva (“OCA”), or Obeticholic acid (INT-747), significantly reduced BC progression and overall tumor burden in a pre-clinical model. The transcriptomic analysis of tumors in mice subjected to OCA treatment revealed differential gene expression patterns compared to vehicle controls. Notably, there was a significant down-regulation of the oncogenic transcription factor MAX (MYC-associated factor X), which interacts with the oncogene MYC. Gene set enrichment analysis (GSEA) further demonstrated a statistically significant downregulation of the Hallmark MYC-related gene set (MYC Target V1) following OCA treatment. In human and murine BC analyses in vitro, agonism of FXR significantly and dose-dependently inhibited proliferation, migration, and viability. In contrast, the synthetic agonism of another common bile acid receptor, the G protein-coupled bile acid receptor TGR5 (GPBAR1) which is mainly activated by secondary bile acids, failed to significantly alter cancer cell dynamics. In conclusion, agonism of FXR by primary bile acid memetic OCA yields potent anti-tumor effects potentially through inhibition of proliferation and migration and reduced cell viability. These findings suggest that FXR is a tumor suppressor gene with a high potential for use in personalized therapeutic strategies for individuals with BC.

## 1. Introduction

Breast cancer (BC) is the most common malignancy and the leading cause of cancer mortality in women. Since the mid-2000s, female BC incidence rates have gradually increased by about 0.5% per year [1]. Microbially-derived metabolites, such as bile acids, may affect cancer progression [2,3]. In fact, it has been demonstrated that BC patients have lower circulating primary bile acids compared to controls [2,4]. We posited that this lack of primary bile acids could be reversed and that activating pathways pharmacologically to mimic higher primary bile acids may be beneficial. To date, the role of microbially derived metabolites in BC has not been well investigated [3]. Therefore, further investigation is needed to understand the influence that bile acid signaling plays on BC progression.

Cholic acid (CA) and chenodeoxycholic acid (CDCA) are two major primary bile acids [2,5]. Primary bile acids can be converted into secondary bile acids through dehydroxylation and deconjugation of the 7α or 7β-hydroxyl groups [2,5]. In humans, key secondary bile acids include lithocholic acid (LCA) and deoxycholic acid (DCA) [2]. Primary bile acids function as signaling molecules primarily through activating the major bile acid receptor (BAR) called farnesoid X receptor (FXR or *NR1H4*), a nuclear receptor [2,5,6,7]. Previous work has suggested an antitumorigenic role for FXR in liver, bladder, and colorectal cancer [8,9,10,11]. Interestingly, we report that BC patients with high FXR expression have greater overall survival uniquely in ER− tumors and the basal-like subtype, but not in the less aggressive luminal A BC subtype or ER+ tumors, suggesting great potential for targeted approaches for subtypes in need of therapeutic advances. Thus, we investigated bile acid receptor signaling using established bile acid receptor mimetic agonists. We demonstrated that agonism of FXR by obeticholic acid (Ocaliva, INT-747, or “OCA”) potently blunted tumor progression in a pre-clinical model. Furthermore, in vitro, OCA significantly blunted cancer cell proliferation and migration, and induced cell death. Interestingly, the agonism of another main BAR, the membrane receptor called G protein-coupled bile acid receptor 1 (*GPBAR1* or TGR5) [2,5] failed to demonstrate anti-cancer effectiveness. Thus, FXR agonism uniquely demonstrated anticancer efficacy. The results presented herein suggest that bile acids or BAR signaling may be beneficial in BC treatment. It is possible that bile acids elevated by bariatric surgery may be one mechanism mediating improved cancer outcomes. Given that OCA is FDA-approved for primary biliary cholangitis (PBC), the incorporation of OCA into studies of cancer risk and as a therapeutic may be a readily translated goal for future clinical studies.

## 2. Materials and Methods

### 2.1. Reagents

All reagents were procured from Sigma-Aldrich (St. Louis, MO, USA), except where specifically mentioned. Obeticholic acid or Ocaliva (“OCA,” INT-747, Cat. No.: HY-12222, MedChemExpress, Monmouth, NJ, USA), TGR5 agonist (“INT-777”, Cat. No.: HY-15677, MedChemExpress) and Paclitaxel (“Ptax”, Cat. No.: N88686, AstaTech, Inc., Bristol, PA, USA; Cat.: HY-B0015, MedChemExpress) were purchased from AstaTech for in vivo work and MedChemExpress for in vitro use. Primary antibodies were purchased from Abcam, Invitrogen, and Proteintech, polyclonal-rabbit GPCR TGR5 (AB72608, Abcam, Waltham, MA, USA), polyclonal-rabbit NR1H4/FXR (AB235094, Abcam), polyclonal-rabbit FXR (PA5-40755, Invitrogen, Waltham, MA, USA), and loading control monoclonal-mouse GAPDH (60004-1-Ig, Proteintech^®^, Rosemont, IL, USA). Appendix A includes all reagents.

### 2.2. Cell Lines and Media

E0771-Luciferase expressing (E0771-luc [12]) and 4T1 cell lines were purchased from ATCC and cultivated in RPMI-1640 medium (Corning, Tewksbury, MA, USA) supplemented with 10% FBS (Gibco, Waltham, MA, USA), 1% penicillin/streptomycin (Pittsburgh, PA, USA), and 1% GlutaMAX (Pittsburgh, PA). Human cancer cell lines, namely MCF7, MDA-MB-231, SUM159 (gift from T. Seagroves), THP-1, and Huh7 (gift from A. Bajwa), were acquired from ATCC (Manassas, VA, USA) unless noted otherwise. All human cancer cell lines were grown in complete DMEM (Gibco) supplemented with 10% FBS, 1% GlutaMAX, 2 mM MEM non-essential amino acids (Gibco), 1 mM sodium pyruvate (Gibco), and 1% penicillin/streptomycin (Gibco). Culturing of the cancer cells was conducted in 100 mm culture dishes under aseptic conditions at 37 °C in the presence of 5% CO_2_. Regular monitoring for Mycoplasma was conducted using the MycoAlert Mycoplasma detection kit (Cat. No.: LT07-318, Lonza, Basel, Switzerland). The cells were passaged when they reached sub-confluent levels, with subculturing limited to the 10th passage for each cell line.

### 2.3. Proliferation Assay

Cell proliferation of human SUM159 and MDA-MB-231 BC cells and murine E0771 BC cells was detected using the IncuCyte S3 live-cell analysis instrument (Sartorius AG, Gottingen, DE, Germany). The SUM159 and MDA-MB-231 cells were seeded into 96-well plates at a density of 1250 cells per well with 100 µL of complete medium. The E0771 cells were seeded into 96-well plates at a density of 1000 cells per well with 100 µL of complete medium. Following an overnight incubation for cell adhesion, 10 µL of FXR or TGR5 agonists (INT-747/OCA and INT-777, respectively) were introduced in a dose-dependent manner, with concentrations of 1 µM, 10 µM, 50 µM, and 100 µM, alongside a positive chemotherapy control, Paclitaxel (Ptax), at indicated concentrations. Cell proliferation measurements were captured every 6 h up to 72 h, assessing cell confluency (%) as the primary parameter. Experiments are representative of *n* = 3–4 biological replicates and include *n* = 4–8 technical replicates. When included, representative images are shown, and videos are available in the Appendix A.

### 2.4. Cell Viability Assay

Cellular viability was assessed using a Vybrant (3-(4,5-dimethylthiazol-2-yl)-2,5-diphenyl-2H-tetrazolium bromide) MTT cell proliferation assay kit (Molecular Probes, Eugene, OR) at 64 h. A total of 12 mM of MTT stock solution was prepared by adding sterile PBS and 10 µL of the solution was added to each well. Plates were incubated at 37 °C for 4 h. A total of 100 µL of SDS-HCl solution was then added to each well and the plates were incubated at 37 °C for another 4 h. Absorbance readings were measured at 570 nm on a Cytation 5 cell imaging multimode reader (BioTek, Winooski, VT, USA). Experiments are representative of *n* = 3 biological replicates and include *n* = 4 technical replicates.

### 2.5. Migration Assay

The migration capability was studied using a wound or scratch assay on the IncuCyte S3 live-cell analysis instrument. MDA-MB-231 and SUM159 triple negative BC (TNBC) cells were plated in 96-well plates at 7500 cells per well, respectively. Once the cells grew to 90% confluence, the scratch was conducted using the IncuCyte wound maker (Sartorius AG). The wells were washed twice with complete growth medium. The cells remained in low serum (1% FBS) and were treated with 0 (DMSO), 10, 50, and 100 µM of FXR agonist OCA or 0 (DMSO), 1, 10, 50, and 100 µM TGR5 agonist INT-777. The plates were analyzed in the IncuCyte S3 Live Cell Imager for 66–78 h. Relative Wound Density (%) was recorded every 6 h until endpoint. Experiments are representative of N = 3 biological replicates and include *n* = 4–8 technical replicates.

### 2.6. Western Immunoblot

Cancer cell lines were cultivated at subconfluency and lysed using RIPA lysis buffer (Cat. No: 20188, EMD Millipore Corp., Burlington, MA, USA) as in previous work [13,14,15]. The cells were subsequently freeze–thawed and vortexed in 1X RIPA lysis buffer, followed by centrifugation at 12,000× *g* RPM for 10 min at room temperature. The protein supernatant was collected, and its concentration was determined using the Rapid Gold BCA protein assay kit (Cat. no: A53225; Pierce^TM^, Appleton, WI, USA). Eighty µg of protein from each cell lysate was loaded onto a 12% SDS-PAGE gel. Protein separation was achieved at 120 Volts for 45 min, and the protein was transferred to a PVDF membrane (Invitrogen) using overnight wet transfer at 4 °C. Immunoblotting was performed using polyclonal anti-rabbit FXR and TGR5 for mouse and human proteins at a ratio of 1:2000 overnight and GAPDH was used as the loading control at a ratio of 1:5000 for 1 h.

### 2.7. Xenograft Model

The UTHSC Animal Care and Use Committee (IACUC) approved all protocols and methods following protocol #23-0432. All animals in the study were kept on a 12 h light/dark cycle and fed a standard breeder’s chow diet. MDA-MB-231 cells were harvested in serum-free DMEM media and suspended into equal volumes of Matrigel (Product No. 354234, Corning^®^, Tewksbury, MA) for orthotopic mammary injection in N = 30 female NSG mice (12–13 weeks old). Five × 10^5^ cells were injected in 100 µL into the 4th right mammary gland at a final Matrigel concentration of 4.5 mg/mL. The injected mice were palpated three times weekly until the tumor grew to a palpable size of 80–150 mm^3^ in volume using a digital caliper. Mice with equivalent tumor size were randomly separated into 3 groups and that included vehicle (40% DMSO and 60% polyethylene glycol (PEG), *n* = 8), OCA (30 mg/kg, *n* = 7), or chemotherapy paclitaxel (“Ptax”, 10 mg/kg, *n* = 6) as a positive control. The animals treated with vehicle or OCA were gavaged daily, and Ptax was administered thrice a week intraperitoneally until the endpoint. Tumor size was calculated using the formula (width^2^ × length/2) [13,14,16]. Body weights were recorded every other day. Mice were sacrificed at endpoint. Tumors were excised for *ex vivo* measurement of tumor volume and weight, photographed, and snap-frozen in liquid nitrogen.

### 2.8. In Silico Analysis

The Kaplan Meier plotter (kmplot.com [17]) database was used to investigate the association between overall survival (OS) and RNAseq candidate gene expression levels. KM-plotter is a real-time publicly available online tool to examine survival analysis using data downloaded (as described by Gyorffy [18]) from the NCBI Gene Expression Omnibus—GEO [19], European Genome-phenome Archive (EGA) [20,21] and TCGA [22,23,24]. KM-plotter uses the HUGO gene nomenclature. We examined NR1H4 (FXR, Kmplot.org RNAseq ID: nr1h4) or MAX (MAX, Kmplot.org RNAseq ID: MAX) expression in this database. We used a follow-up threshold of “all” which included all follow-up months available. For restrictions to ER+, ER−, or the PAM50 subtypes, either the most prevalent BC subtype luminal A or the most aggressive TNBC basal-like subtype were used, and the sample sizes were 2575, 214, 1504, and 309, respectively, which are sample data available in the publicly accessible website kmplot.com. We did not select or restrict to endocrine or chemotherapy treated. We did not select for trichotomization. We did not modify the standard selections, such as cutoff, and left the selection checked for “auto select best cutoff.” When selecting for auto cutoff, the best-performing threshold is used after the computation of all possible cutoffs between lower and upper quartiles [25]. Per the KMplot website describing the best cutoff, the Benjamini–Hochberg analysis was used to generate the false discovery rate (FDR) to correct for multiple hypothesis testing. The cutoff value with the highest significance (lowest FDR) was determined for *NR1H4* and the cutoff values were as follows: ER+ and ER− = −3.0; luminal A = −2.94; and basal-like = −2.61. For *MAX*, the cutoff values were as follows: ER+ = 5.54; ER− = 5.08; luminal A = 5.54; and basal-like = 4.99. The hazard ratios (HRs) with 95% confidence intervals and log-rank *p*-values are shown in the figures. The red lines demonstrate high levels and the black lines demonstrate low levels of expression using the RNAseq database for breast cancer patients.

### 2.9. RNA-Seq and Pathway Analysis of MDA-MB-231 Tumors

Total RNA was isolated from tumors using the RNeasy Mini Kit (#74104, Qiagen) according to the manufacturer’s instructions for the vehicle controls (*n* = 7) and the OCA-treated (*n* = 6) fresh-frozen samples. The integrity of the RNA was assessed using Agilent TapeStation, and samples with RIN > 6.0 were used. mRNA-seq libraries for the Illumina platform were generated and sequenced at Azenta using the Illumina HiSeq 2 × 150 bp configuration following the manufacturer’s protocol.

Fastq files were obtained from Illumina HiSeq, underwent quality control assessment via FastQC, and were first aligned to the human reference genome build version hg38 (Gencode v45) using STAR [26]. Aligned reads were sorted by SAMtools [27] and sorted reads were subjected to the quantification via Salmon [28]. ENSENBL gene-level counts were used for data analysis in R version 4.3.1 [29]. Read counts loaded from salmon quant files via tximport [30] were subjected to differential gene expression analysis between the vehicle or OCA treatment groups via DESeq2 [31]. An adjusted *p*-value < 0.1 was used to determine differentially expressed genes (DEGs) from each sample group. Read counts were normalized for downstream analyses and visualization using the regularized log normalization (rlog) from DESeq2. For clustering analysis, the ComplexHeatmap R package [32] was used to represent normalized and scaled gene expression values in heatmaps where rows (genes) were clustered via complete linkage methods and columns (samples) were split by groups in the context of semi-supervised clustering.

The significantly upregulated and downregulated genes were submitted to the web-based server DAVID [33,34] for gene ontology and pathway analysis. For volcano plots, differentially expressed genes with adjusted *p*-values less than 0.1 were plotted, and the colors were applied based on the significance criteria (adjusted *p*-value ≤ 0.05 and log2|FC| ≥ 0.58).

### 2.10. Gene Set Enrichment Analysis (GSEA) of MDA-MB-231 Tumor RNA

GSEA software version 4.3.3 was used for the identification of enriched gene signatures [35]. GSEA analysis was performed by using the rlog-normalized gene expression data obtained from 7 vehicle-treated and 6 OCA-treated tumor samples. A total of 1000 gene set permutation parameters were used to test for significance at a false discovery rate (FDR) threshold of 0.25 (25%). MSigDB Human Collection hallmark gene sets [36] were used to determine enriched pathways in the vehicle vs. OCA groups. For GSEA hallmark gene sets, nominal *p*-values of less than 0.05 were shown with their enrichment plots.

### 2.11. Statistical Analysis

Values were presented using GraphPad Prism 10.0.2 as the mean ± standard error of the mean (SEM) from at least three independent biological replicate experiments, as noted above. Significance was assessed by Student’s *t*-test or the one-way analysis of variance (ANOVA) method in GraphPad Prism and is represented with asterisks (*) in the figures, indicated as *p* > 0.05 *, *p* > 0.01 **, *p* > 0.001 ***, and *p* > 0.0001 ****.

## 3. Results

### 3.1. Higher NR1H4 Expression Is Associated with Greater Survival in Patients with ER− and Basal-like BC Subtypes

Overall survival was analyzed in KMPlotter using the RNAseq data [25]. Expression of FXR (*NR1H4*) did not alter overall survival in patients with ER+ tumors (Figure 1A). Likewise, greater *NR1H4* expression in the luminal A subtype, the most prevalent and least aggressive subtype, reduced survival slightly (Figure 1B). However, in patients with ER− tumors, higher expression of *NR1H4* was associated with improved overall survival (Figure 1C) with similar protection in patients with tumors of the basal-like subtype, typically TNBC (Figure 1D), although these findings were not significant. Taken together, in patients with the most aggressive subtypes, including ER− and basal-like BC, greater expression of *NR1H4* appeared to improve survival.

### 3.2. FXR Is Expressed in Human and Murine Triple-Negative Breast Cancer Cell Lines

Because patients with high levels of expression of NR1H4 demonstrated greater survival in the ER− and basal-like subtypes, which are often TNBC, we aimed to investigate the effects of FXR agonism on TNBC cell lines. We first determined the protein expression of FXR in multiple cell lines. Western immunoblot analysis demonstrated the presence of FXR in human BC cell lines (MDA-MB-231, MCF7, and SUM159). Quantification of triplicate blots demonstrated particularly high levels of FXR protein in MDA-MB-231 (even after high passage) and MCF7 cells, with the lowest levels in SUM159 cells (Figure 2A,B). THP1 cells, a monocytic cell line that expresses low FXR, were included as a negative control, while Huh7 cells, a hepatoma-derived hepatocyte line that should express FXR, were evaluated as a positive control. Murine E0771 and 4T1 cell lines also express FXR (Appendix A). In summary, the evidence suggests high levels of FXR conserved across multiple BC cell lines.

### 3.3. OCA Treatment Reduced TNBC Tumor Growth in Xenograft Model

Next, to determine if the agonism of FXR could impact tumor progression, we examined OCA treatment in a TNBC xenograft model. Female NSG mice were injected with MDA-MB-231 cells, which resulted in the development of palpable tumors after 36 days. When the tumors reached 80 mm^3^, the mice were randomized based on tumor size to treatment groups. The vehicle alone or the OCA FXR agonist was administered orally once daily at a dose of 30 mg/kg in a vehicle composed of 40% DMSO and 60% PEG. In a third cohort, paclitaxel (Ptax) was injected intraperitoneally three times a week at a dose of 10 mg/kg as a positive control intervention. Body weights were not impacted by vehicle, OCA, or Ptax. OCA treatment significantly reduced breast tumor progression (Figure 3A) compared to the vehicle group. Ptax reduced tumor progression as expected. The experimental endpoint analyses demonstrated a significant 2.2-fold reduction in tumor volume and 2.6-fold reduction in tumor mass with OCA treatment compared to the vehicle controls (Figure 3B,C). Representative images of excised tumors are shown in Figure 3D. These findings highlight OCA as a potent therapeutic for inhibiting breast tumor growth in vivo.

### 3.4. Transcriptomic and GSEA Analysis of OCA-Treated TNBC Tumors Reveal Downregulation of Hallmark MYC Target Gene Set

RNA was isolated from the fresh-frozen tumors to examine the transcriptional pathways responsible for the reduced tumor burden in the OCA-treated mice. Genes regulated by OCA compared to the vehicle are shown in the heatmap in Figure 3E and the volcano plot in Figure 3F. OCA increased the expression of genes such as DST (dystonin), which is a tumor suppressor in breast cancer that regulates cell adhesion [37]. POGZ (pogo transposable element derived with ZNF domain), which controls mitotic fidelity and genome stability [38], is also increased in response to OCA. In contrast, OCA downregulated the basic helix-loop-helix leucine zipper (bHLHZ) transcription factor MAX (MYC-associated factor X) which has been shown to bind to the oncogene MYC. Elevated MAX expression levels have been observed to correlate with breast cancer cell proliferation, glycolytic activity, and migratory potential [39]. High expression of MAX in breast cancer patients with ER+ or luminal A subtype of ER+ tumors displayed significantly improved overall survival (Appendix A, *p* = 0.0011 and *p* = 0.0062, respectively) while a high expression of MAX in ER− or basal-like TNBC subtype displayed significantly worse survival over time (Appendix A, *p* = 0.008 and *p* = 0.014, respectively). These data suggest that the OCA-mediated downregulation of MAX in TNBC tumors will improve survival. Importantly, GSEA demonstrated a significantly enriched Hallmark MYC Target V1 gene set in the vehicle-treated tumors with downregulation after OCA treatment (Figure 3G, nominal *p*-value 0.025).

### 3.5. Agonism of FXR Reduces Triple-Negative Breast Cancer Cell Proliferation, Viability, and Migration

To investigate the impacts of OCA on cancer cells, tests including cell proliferation, viability, and migration assays were performed in vitro by activating FXR over time at concentrations of 0–100 µM. Treatment with OCA demonstrated a potent dose-dependent decrease in MDA-MB-231 BC cell proliferation. Cell proliferation was effectively decreased in OCA-treated cells with doses of 50 and 100 µM, with minimal inhibition quantified with 10 µM (Figure 4A). Similarly, a consistent reduction in cell viability was measured with higher doses compared to the DMSO control and lower doses of OCA and was comparable to the Ptax positive control (Figure 4B). Cell migration is commonly measured using the wound or scratch assay as a marker of aggressive cancer subtypes to approximate metastasis. Again, migration of MDA-MB-231 cells was not impaired at the lower dose of 10 µM but was significantly reduced in the cells treated with 50 and 100 µM (Figure 4C,D, and videos in Appendix A). Analysis of another human TNBC cell line, SUM159, showed identical dose- and time-dependent results on proliferation, viability, and migration, with 100 µM OCA having the most potent impacts (Appendix A). To determine if OCA’s anti-cancer effects were conserved, the murine TNBC cell line E0771 was examined for effects on cell proliferation, viability, and migration. Similar to findings in human cell lines, OCA dose-dependently blunted proliferation and viability, with 100 µM leading to excessive cell death (Appendix A). Migration was also significantly inhibited at 50 µM compared to DMSO controls. Overall, agonism of FXR using a synthetic bile acid mimetic appears to be an effective anti-cancer therapy through reduced cancer cell viability and impaired proliferation and migration.

### 3.6. TGR5 Bile Acid Receptor Agonism Failed to Reduce Triple-Negative Breast Cancer Cell Proliferation, Viability, and Migration

Lastly, to determine if the anti-cancer cell activity was unique to FXR agonism, INT-777, a synthetic agonist of the bile acid receptor TGR5, was also evaluated for anti-cancer effectiveness. TGR5 is highly expressed in human BC cells. Appendix A demonstrates that in human MDA-MB-231 and SUM159 cell lines, TGR5 agonism failed to demonstrate anti-cancer efficacy on cell proliferation, viability, or migration (Appendix A). Similarly, in murine E0771 cells, INT-777 failed to blunt progression or viability, with a negligible impact on migration (Appendix A). In fact, in some cases, high doses of INT-777 appeared to increase proliferation and migration in human cells above the DMSO levels.

## 4. Discussion

Bile acids have been shown to regulate cancer progression or metastasis in other cancers, but prior to this study, BA-mediated mechanisms had not yet been extensively explored in TNBC [3]. The reduction of BC risk after surgery [40] is correlated with elevated bile acid pools observed after bariatric surgery [41], thus revealing possible therapeutic avenues for BC patients related to bile acid signaling. We therefore took advantage of commercially available, and in the case of OCA, FDA-approved, synthetic bile acid mimetics. The key findings reported in this study demonstrate a role of the bile aid receptor FXR in BC progression. Our data demonstrated the following: (i) patients with higher levels of FXR appear to have greater survival, but only in ER− and basal-like subtypes, not ER+ or luminal A subtype; (ii) human and murine BC cell lines express FXR; (iii) the FXR agonist OCA slowed tumor progression in a murine model, potentially through downregulation of the MYC pathway leading to decreased expression of the oncogenic transcription factor *MAX*; (iv) in vitro OCA demonstrated potent dose-dependent inhibition of proliferation and migration, as well as reduced viability. These findings appear unique to FXR agonism because the TGR5 agonist INT-777 minimally reduced cell growth and viability and did not impact cell migration. Indeed, others have also reported that FXR expression is associated with indicators of better outcomes in BC and is an independent prognostic indicator of survival [42,43], but these studies did not focus on a subtype-specific effect. Girisa et al. reviewed the role of FXR in multiple cancers [44]. In breast cancer, citing only in vitro studies, varied responses to agonism were evident in studies using mostly the GW4064 compound (which some report is non-specific [45]), bile acid chenodeoxycholic acid (CDCA), or the plant extract guggulsterone. Interestingly, no reports of OCA (INT-747) were included. The striking ER− survival advantage in patients points to the potential for enhanced personalized medicine using BAR agonists. FXR is detected in benign and malignant breast tissue, and there it is associated with increased apoptosis [46]. FXR induces apoptosis in normal and malignant BC cells in vitro [47,48,49], which is in line with our findings. Taken together, agonism of the bile acid receptor FXR may be beneficial to patient survival through direct effects on cancer cell survival and migration. We posited that elevated bile acid metabolites may increase FXR activation potentially through increased apoptosis, reducing breast neoplastic progression.

Microbes may play a vital role in cancer biology considering that diversity of the microbiome can be altered by both obesity and cancer, although there is come controversy in the field regarding [3,50,51]. Microbes can modify levels of their own metabolites and host metabolites, including bile acids [2]. Primary (1°) and microbially modified (secondary, 2°) bile acids are not just important in digestion but are bioactive mediators that may signal as paracrine or endocrine factors. Primary bile acids, including cholic acid (cholate, CA), chenodeoxycholic acid (CDCA), and muricholic acid (MCA, in mice), are synthesized from cholesterol via classic and alternative pathways in the liver [2]. Primary bile acids are conjugated with glycine (GCA, GCDCA) or taurine (TCA, TCDCA) in the liver before being secreted into the intestine. Certain microbes rich in 7-alpha-hydroxylase, especially *Clostridium cluster XIVa* and *Bacteroides*, deconjugate 1° bile acids to allow for 1° to 2° bile acid conversion, which is facilitated by a wide variety of additional microbes. Bile acids may function as potent signaling hormones that regulate energy expenditure, inflammation, and cancer, in part through the activation of transmembrane or nuclear hormone bile acid receptors. Patients with BC have been reported to have reduced levels of bile acids [5]. Interestingly, the presence of bile acids in breast tissue was identified decades ago in humans, with evidence that oral bile acid administration preferentially accumulated in the breast [52,53]. Cook et al. identified bile acids in non-human primates that were modified by Western or Mediterranean diet, and these altered bile acids were importantly not reflected in circulating bile acid pools [54]. These data point to exciting future studies in the increasingly complex roles of microbes and microbially derived metabolites in cancer risk and progression.

Patients and pre-clinical models that are obese display microbial alterations compared to lean controls. Obesity is associated with various types of cancer, including elevated BC risk and mortality [3,55]. Two-thirds of US adults are overweight or obese, with a higher prevalence in women and disproportionately higher rates among minorities [56]. The World Health Organization (WHO) defines obesity as having a Body Mass Index (BMI) of more than 30 kg/m^2^ [57]. The most effective treatment option for obese patients today is bariatric surgery [55]. Recent findings have demonstrated that weight loss by bariatric surgery is not only protective, especially for ER− tumors including TNBC [3,40], but also improves clinical outcomes [56]. Furthermore, we recently demonstrated that weight loss by bariatric surgery decreased TNBC tumor progression in murine models compared to obese controls receiving sham surgery [14]. It is possible that bile acids elevated by bariatric surgery may be one mechanism by which improved cancer outcomes are observed in bariatric surgery patients even 5–10 years post-surgery. Obesity and weight loss have potent effects on the gut microbiome. Though mechanisms underlying bariatric surgery-associated protections are unknown, increasing interest in the field of microbially derived metabolites suggests that cancer could be mediated in part by metabolites such as bile acids and/or agonism of BARs, as indicated by the work presented herein.

BC subtypes are diagnosed by the presence or absence of estrogen receptor (ER), progesterone receptor (PR), and human epidermal growth factor receptor 2 (HER2), and tumors that lack all three receptors are triple-negative BC (TNBC) [2]. TNBC is a highly aggressive subtype and typically cannot be treated with targeted approaches because most therapies on the market target these three receptors. Considering the limited targeting therapies, patients with TNBC have higher rates of recurrence and metastasis, with poorer overall survival than those with other BC subtypes [58,59]. This gap in clinical care emphasizes the importance of discovering new targeted cancer prevention strategies for TNBC to improve patient survival, such as FXR agonism. An analysis of immunotherapy responsiveness across multiple cancers [60] showed that patients with high FXR (*NR1H4*) performed significantly better, with higher overall survival after anti-PD-L1 therapy. However, it should be noted that breast cancer is not yet available in that database [60]. While chemotherapy is the standard treatment option for TNBC, immunotherapies have recently been approved. Clinically, not every patient responds well to immunotherapy or may have adverse reactions [61]. As more TNBC patients receive immunotherapy, it will be interesting to investigate the interaction of bile acids, FXR expression, mutations, or other factors that may mediate response to therapy.

## 5. Conclusions

In summary, findings suggest that FXR functions as a tumor suppressor. FXR agonism drastically reduced aggressive cellular cancer dynamics and in vivo demonstrated reduced tumor progression. Taken together, our data provide important support for the potential role of BAR FXR agonism in BC as a potential therapeutic with a tumor suppressor function.

## Figures and Tables

**Figure 1 cancers-16-01368-f001:**
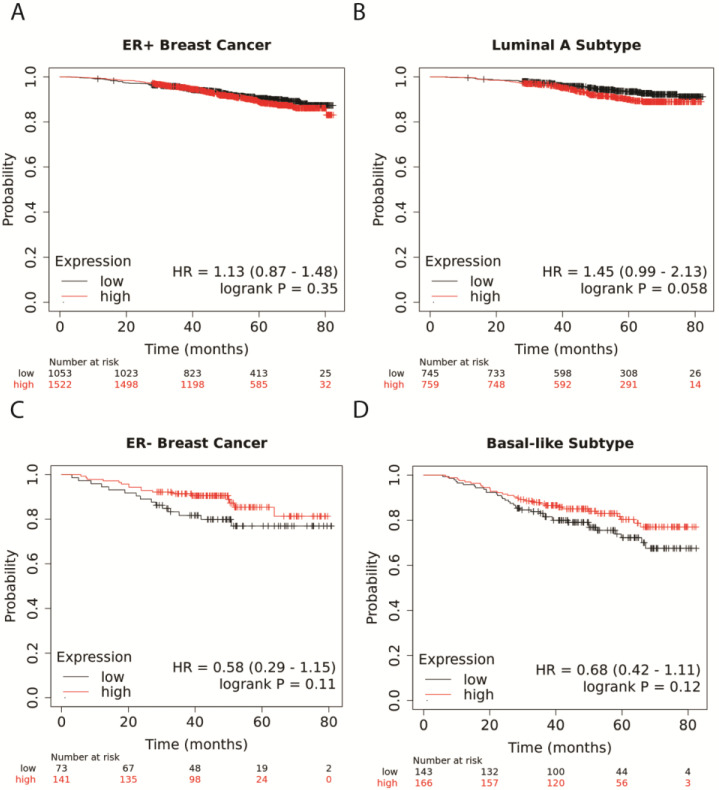
Higher NR1H4 expression is associated with greater survival in patients with ER− and basal-like breast cancer subtypes. Overall survival was analyzed in KMPlotter using RNAseq data in estrogen receptor-positive (ER+) breast cancer (**A**), luminal A ER+ breast cancer subtype (**B**), ER− breast cancer (**C**), and basal-like TNBC breast cancer subtype (**D**). The number of patients at the indicated time in each month is shown below each graph for low and high expressors. High expression of NR1H4 (red) is compared to low expression (black) with hazard ratios (HR) and log-rank P.

**Figure 2 cancers-16-01368-f002:**
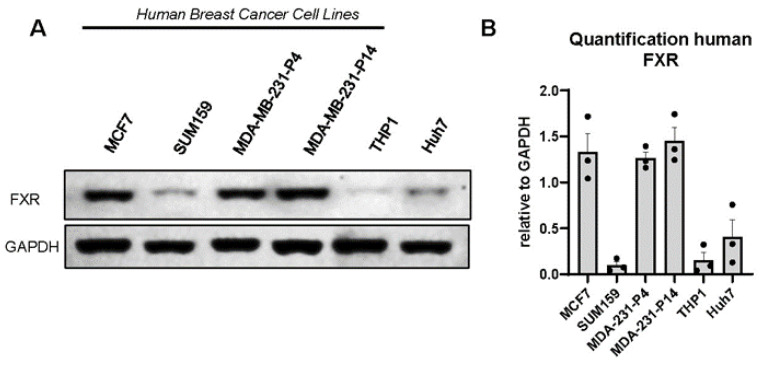
The bile acid receptor FXR is expressed in human breast cancer cell lines. (**A**) Endogenous FXR was detected in MCF7 (ER+ cell line), SUM159, and MDA-MB-231 triple-negative breast cancer (TNBC) cell lines (56 kDa). MDA-MB-231, at passage (“P”) 4 and passage 14, were compared to demonstrate that, with high passage, FXR is not lost. For our studies, we use passages <P10. Human monocytic THP-1 and hepatocyte Huh7 cell lines were used as negative and positive controls for FXR, respectively. GAPDH was used as a loading control (37 kDa). Representative images are shown from *n* = 3 Western immunoblots. (**B**) Images were quantified using ImageJ with the relative intensity of protein expression of FXR normalized to GAPDH loading control with mean −/+ SEM shown.

**Figure 3 cancers-16-01368-f003:**
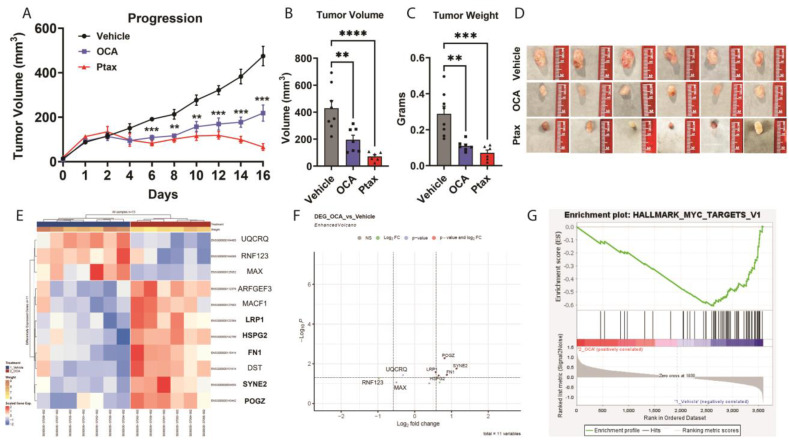
FXR agonist OCA reduced TNBC tumor burden. Age-matched female NSG mice were injected with MDA-MB-231 human TNBC cells. When tumors reached 80 mm^3^, mice were randomized based on tumor size, and pharmaceutical interventions were initiated. Vehicle or OCA was gavaged daily at a dose of 30 mg/kg in 40% DMSO/60% PEG vehicle. Intraperitoneal injections of paclitaxel (Ptax, 10 mg/kg) were completed three times a week as a positive control. Tumor size was measured by a digital caliper daily at indicated times. (**A**) Tumor progression was measured for 16 days. (**B**,**C**) Tumor volume at the endpoint and excised mass were quantified. (**D**) Representative images of excised tumors from each treatment group are shown. (**E**) Semi-supervised heatmap of RNAseq transcriptomic analysis of tumors from vehicle- or OCA-treated mice. Treatment group and tumor weight are shown below the dendrogram. Gene names at right of heatmap are bolded to reflect significance shown in (**F**) volcano plot. (**G**) Gene Set Enrichment Analysis (GSEA) enrichment plot for Hallmark_MYC_Targets_V1 with nominal *p*-value 0.025. Data are presented as mean ± SEM. *n* = 6–8 mice per group. The statistical significance was determined using the one-way or two-way ANOVA based on LSD- generated using GraphPad Prism 10.1. Significance levels are denoted as ** *p* > 0.01, *** *p* > 0.001, and **** *p* > 0.0001.

**Figure 4 cancers-16-01368-f004:**
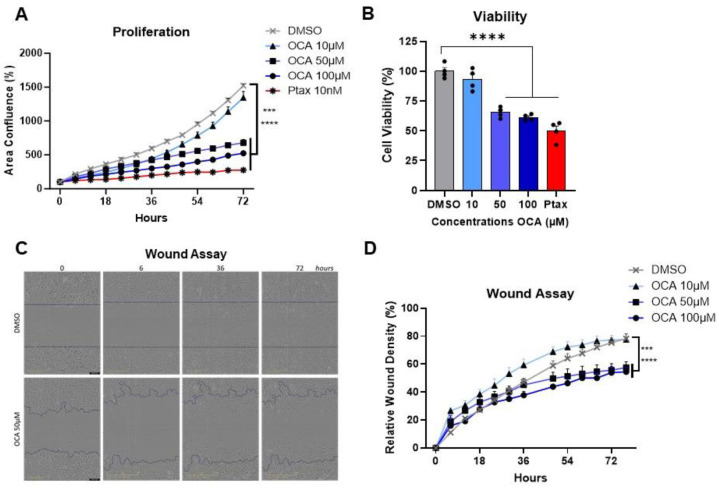
Impact of FXR agonism on cell proliferation, viability, and migration. (**A**,**B**) MDA-MB-231 TNBC cell lines were plated in 96 well plates at 1250 cells per well and IncuCyte Live Cell Imager quantified data at the indicated time-points. Cells were treated with increasing concentrations (0–100 µM) of obeticholic acid (OCA, INT-747). Paclitaxel (Ptax) was used as a positive control at 10 nM. (**A**) Proliferation was measured as area confluence (%). (**B**) Cell viability was quantified by MTT assay at 64 h time-point. (**C**,**D**) Cell migration was quantified using the IncuCyte by a migration wound or scratch assay. MDA-MB-231 cells were plated in 96 well plates at 7500 cells per well. (**C**) Representative images at time 0, 6, 36, and 72 *h* are shown for DMSO and 50 µM OCA. (**D**) Quantification of relative wound density (%). Videos of the wound closure are available in Appendix A: SV1 for DMSO treated and SV2 for 50 µM OCA. Statistical analysis was conducted using Fisher’s LSD or one-way ANOVA. Experiments include *n* = 4 technical replicates and are representative of *n* = 3 biological replicates. Data are shown as means −/+ SEM. *** *p* < 0.001, **** *p* < 0.0001.

## Data Availability

The original contributions presented in the study are included in the Appendix A, further inquiries can be directed to the corresponding authors.

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
