# Peer review of "FXR Agonism with Bile Acid Mimetic Reduces Pre-Clinical Triple-Negative Breast Cancer Burden"

_cancers, 2024, doi:10.3390/cancers16071368_

Round 1

Reviewer 1 Report

Comments and Suggestions for Authors

This study underscores the therapeutic potential of FXR, a bile acid receptor, especially in patients with estrogen receptor-negative (ER-) breast cancer (BC), linking higher FXR expression to enhanced survival rates. Activation of FXR by FDA-approved agonists has been shown to significantly curtail BC progression and tumor burden in pre-clinical models and to inhibit BC cell proliferation, migration, and viability in vitro, a response not mirrored by the activation of GPBAR1 (TGR5), a secondary bile acid receptor. While the study is methodically sound and offers valuable insights into FXR's role in BC, it requires further exploration to strengthen its findings:

1. The tumor suppressive function of FXR has reported to depend on hepatic modulation of bile acid homeostasis. Given that the current study presents a breast cancer cell autonomous function, the author should provide at least some mechanistic insights. Are there relevant pathways altered? Are there any FXR targets altered that may contribute the phenotype?

2. In Figure 2, the authors showed different levels of FXR in breast cancer cell-lines. To further investigate the effect of FXR, the authors should consider knocking-out FXR in FXR high cells and assessing the tumor cell phenotypes.

3. In Figure 1D, HR = 0.66 (0.37 – 1.17), P = 0.15, these statistics do not support a significant difference in basal like breast cancer.

Reviewer 2 Report

Comments and Suggestions for Authors

Makowski, Pierre and colleagues investigated the effects of FXR agonism by obeticholic acid (OCA) in pre-clinical models of Triple Negative Breast Cancer (TNBC). Their main findings suggest that FXR agonism reduces proliferation, viability and migration of TNBC MDA-MB-231 cell line (an activity unique to FXR agonism because TGR5 agonism does not have the same effect) and reduced TNBC tumour growth in a xenograft model.

FXR may have a tumour promoting or a tumour suppressor role, depending on cancer type and/or histological-molecular cancer subtype. This study provides convincing evidence of a tumour suppressor role of FXR in TNBC. Although the current work provides little mechanistic evidence of how FXR mediates this tumour suppressor activity in TNBC cells, the main findings of the authors are of importance for the breast cancer oncology field. However, there are issues that need to be addressed before publication

Major points

- Figure 1 and paragraph 2.8 a)The authors used KM plot to generate the survival curves. The cohort (is it the TCGA-breast cancer cohort?) and the  parameters of KM plot analysis should be described in detail (especially the cut off that was selected to stratify patients, the probe and the subgroups). I tried the same analysis and got different results because I cannot know the parameters used by the authors. b) Why the x-axis of the KM curves is restricted to 40-50 months? KM plotter and other similar software generate KM curves with a timeline of 120+ months for breast cancer patients (the same is possible for NR1H4 curves in KM plotter) c) The data should be interpreted with caution because the generated p-values are not adjusted for multiple hypothesis testing, as mentioned in the KM plotter website. The authors should discuss this and may calculate the False Discovery Rate or use a Benjamini-Hochberg correction method to adjust the p-values.

- Figure 2. A. From the quantification panel, it seems that the expression of FXR in the SUM159 TNBC cell line is much lower than in the MDA-231 and similar to the negative control. Based on this, it may not be accurate to include SUM159 in the list of BC cell lines that express FXR. 1) What can explain the difference between FXR expression in SUM159 and MDA-231 cells 2) Do the different levels of expression explain the different effect of OCA in these cells (Figures 4 and S2)?

B. Is there any difference between the expression of FXR in normal breast tissue and breast cancer or between breast cancer subtypes?

- There are a few studies reporting tumour promoting or tumour suppressor activity of FXR in BC. These are summarised in PMID: 35006466 and should be discussed in the manuscript (especially the studies performed on TNBC cells)

Comments on the Quality of English Language

A well written manuscript that needs minor editing for typos.

Round 2

Reviewer 1 Report

Comments and Suggestions for Authors

No further comments

Reviewer 2 Report

Comments and Suggestions for Authors

The authors addressed my concerns in a satisfactory way. The revised manuscript is significantly improved and can be published.